# Extracellular Overexpression of a Neutral Pullulanase in *Bacillus subtilis* through Multiple Copy Genome Integration and Atypical Secretion Pathway Enhancement

**DOI:** 10.3390/bioengineering11070661

**Published:** 2024-06-28

**Authors:** Wenkang Dong, Xiaoping Fu, Dasen Zhou, Jia Teng, Jun Yang, Jie Zhen, Xingya Zhao, Yihan Liu, Hongchen Zheng, Wenqin Bai

**Affiliations:** 1Key Laboratory of Industrial Fermentation Microbiology, Ministry of Education, The College of Biotechnology, Tianjin University of Science and Technology, Tianjin 300457, Chinalyh@tust.edu.cn (Y.L.); 2National Technology Innovation Center of Synthetic Biology, Tianjin Institute of Industrial Biotechnology, Chinese Academy of Sciences, Tianjin 300308, China; 3Industrial Enzymes National Engineering Research Center, Tianjin Institute of Industrial Biotechnology, Chinese Academy of Sciences, Tianjin 300308, China; 4Tianjin Key Laboratory for Industrial Biological Systems and Bioprocessing Engineering, Tianjin Institute of Industrial Biotechnology, Chinese Academy of Sciences, Tianjin 300308, China; 5Key Laboratory of Engineering Biology for Low-Carbon Manufacturing, Tianjin Institute of Industrial Biotechnology, Chinese Academy of Sciences, Tianjin 300308, China

**Keywords:** *Bacillus subtilis*, pullulanase, atypical secretion, large molecular transporters, trehalose

## Abstract

Neutral pullulanases, having a good application prospect in trehalose production, showed a limited expression level. In order to address this issue, two approaches were utilized to enhance the yield of a new neutral pullulanase variant (PulA3E) in *B. subtilis*. One involved using multiple copies of genome integration to increase its expression level and fermentation stability. The other focused on enhancing the PulA-type atypical secretion pathway to further improve the secretory expression of PulA3E. Several strains with different numbers of genome integrations, ranging from one to four copies, were constructed. The four-copy genome integration strain PD showed the highest extracellular pullulanase activity. Additionally, the integration sites *ytxE*, *ytrF*, and *trpP* were selected based on their ability to enhance the PulA-type atypical secretion pathway. Furthermore, overexpressing the predicated regulatory genes *comEA* and *yvbW* of the PulA-type atypical secretion pathway in PD further improved its extracellular expression. Three-liter fermenter scale-up production of PD and PD-ARY yielded extracellular pullulanase activity of 1767.1 U/mL at 54 h and 2465.1 U/mL at 78 h, respectively. Finally, supplementing PulA3E with 40 U/g maltodextrin in the multi-enzyme catalyzed system resulted in the highest trehalose production of 166 g/L and the substrate conversion rate of 83%, indicating its potential for industrial application.

## 1. Introduction

Pullulanase (EC 3.2.1.41), as a kind of starch debranching enzyme, specifically hydrolyzed α-1, 6-glucosidic linkages, plays an important application role in the starch processing industry [1,2,3]. Pullulanase could improve the utilization of raw materials and reduce costs in the starch processing industry to produce glucose, maltose, trehalose, modified starch, and other products [4,5,6,7]. Except the application in saccharification processing needs acidic pullulanase, there are still so many industries, such as the production of trehalose and modified starch also need neutral pullulanase in the enzymatic treatment [8,9,10,11]. In the current industrial production of trehalose, the addition of pullulanase through the two-enzyme method is commonly practiced to enhance the conversion rate of trehalose from starch or maltodextrins. However, it is important to note that commercial pullulanases typically possess an acidic nature, rendering them unsuitable for the optimum reaction pH required by the neutral maltooligosyltrehalose synthase (MTSase) and maltooligosyltrehalose trehalohydrolase (MTHase) that are widely employed in the current industrial production of trehalose [8,12,13]. Additionally, there is currently a lack of neutral pullulanase products on the market, and relatively few neutral pullulanases have been reported [8]. It is, therefore, crucial to look for innovative neutral pullulanases and construct a highly productive neutral pullulanase-producing expression system because of the rising demand for this enzyme. So far, there are not many reports on overexpression and application of neutral pullulanases [1,5].

*Bacillus* strains such as *B. subtilis* have been widely used in the industrial production of recombinant proteins for their characteristics of being generally recognized as safe (GRAS) and owning powerful secretion capability [14,15,16,17]. During the last few decades, much research on host strain modification, promoter and signal peptide optimization, and fermentation optimization has been performed to improve the production level of heterologous proteins in *B. subtilis* [18,19,20]. However, there is not a commonly valid strategy for enhancing the expression of various heterologous proteins. The protein secretion mechanisms of B. subtilis are a focal point in industrial settings and necessitate additional research and enhancement to facilitate the production of various specific proteins [21,22,23]. Though the Sec pathway is the main protein secretion pathway in *B. subtilis*, few heterologous proteins have successfully resulted in high-level secretory expressions using the typical Sec signal peptides [24]. Numerous studies have been conducted thus far on the applications of signal peptides in *B. subtilis* [25,26]. However, limited efforts have been made to uncover the self-secretory mechanism of pullulanases with atypical secretion signal peptides [27].

However, in our earlier research, we were the first to identify an *Anoxybacillus* sp. pullulanase (PulA) that was expressed secretively through an atypical secretion pathway in *B. subtilis* [27]. Based on the findings from our prior research, both the neutral pullulanase PulA and its mutant R503E/I506E/H507E (PulA3E) with high specific activities at 60 °C and pH 6.0 and high thermostability above 60 °C showed a good potential in the industrial application [8,27]. Therefore, this work focuses on the construction of strains suitable for industrial production by multiple copies genome integration to overcome the instability of fermentation-producing of the expressing strains relying on free plasmids. Meanwhile, the novel regulation theory for the atypical secretion pathway revealed in our previous work has been verified and applied in this study to enhance the secretory expression of the target pullulanase. At last, the effectiveness of scaling up fermentation in a 5 L fermenter and utilizing the industrial production strain of neutral pullulanase (PulA3E) for enzymatic trehalose preparation was assessed following its successful construction.

## 2. Materials and Methods

### 2.1. Bacterial Strains, Plasmids, and Chemicals

In our previous study, we constructed *B. subtilis* 0127, a strain that enhances atypical secretion by silencing the encoding gene (*ytxE*) of a macromolecular protein transporter in *B. subtilis* SCK6 [21]. This strain has been deposited in our laboratory as the original expression host in this study. For the production of neutral pullulanase, we employed the gene *pulA3E*, encoding the triple-site mutants R503E/I506E/H507E of PulA, which has been deposited in NCBI under the accession number GenBank: AEW23439.1 [28]. Thermo Fisher Scientific Co., Ltd. (Waltham, MA, USA) provided the restriction endonucleases and DNA polymerase, while TaKaRa Biotechnology (Dalian, China) Co., Ltd. supplied all other enzymes, chemicals, and reagents for the experiment.

### 2.2. Construction of the Genome Integration Strains

The integration of the target genes into the genome was carried out through the utilization of the CRISPR/Cas9 system established in a prior study, which involved the plasmids pHT43-cas9 and pUC980-2-gRNA-temp [23]. The plasmids used for genome integration of *pulA3E* were listed in Appendix A. These plasmids were fabricated through a Golden Gate assembly reaction and subsequently introduced into *B. subtilis* 0127 using the highly proficient method of transformation [29]. After that, the positive recombinant strains were selected on LB agar plates supplemented with chloramphenicol (10 µg/mL) at 30 °C. The strain editing was induced at a temperature of 30 °C by adding 0.1 mM IPTG to the culture broth, which was then diluted to an initial OD600 nm of 0.1 in 3 mL of LB medium containing 10 µg/mL chloramphenicol. This cultivation process lasted for 24 h. In order to confirm the occurrence of the desired genetic modifications, the genomic DNAs of specific colonies were extracted and subjected to PCR amplification at the targeted locus using the appropriate primers (Appendix A). The correctly edited strains were incubated in 1 mL LB medium supplemented with 0.0005% SDS at 37 °C, 220 rpm for 12 h. Thereafter, the precipitated cells obtained by centrifugation were then transferred into fresh LB medium. After 5 h cultivation at 50 °C, 220 rpm, the culture was spread on LB plates with and without 10 µg/mL chloramphenicol separately with gradient dilutions (10^3^–10^7^) to select the strains eliminating plasmids.

### 2.3. Overexpression of the Encoding Genes of the Key Transporters

The amplified gene fragments (ytnA, comEA, spoIIQ, and yvbW) were ligated with the linearized plasmid pMA05 using the primers listed in Appendix A. This ligation process was carried out with a seamless cloning kit from GenScript Co., Ltd., located in Nanjing, China, following the instructions provided by the manufacturer [21]. Thereafter, the recombinant plasmids were transformed into the competent cells of *B. subtilis* PD using the procedure outlined in our prior research [21]. The positive recombinant colonies were selected on LB agar plates supplemented with 25 µg/mL kanamycin at 37 °C for 12–16 h. The recombinant strains were verified by bacterial cell PCR with the primers ytnA-F/R, comEA-F/R, spoIIQ-F/R, and yvbW-F/R in Appendix A. The positive colonies were transferred to fresh LB culture liquid (25 µg/mL kanamycin) for a 12 h cultivation period. Following this, the culture broth was moved to SR medium with a 1% inoculum for shake flask cultivation at 37 °C and 220 rpm for a total of 120 h.

### 2.4. Fermentation of the Recombinant Strains

For the primary seed cultivation, 50 μL glycerin tube frozen culture was inoculated into 5 mL LB medium and was incubated at 37 °C and 220 rpm for 12 h. Moreover, 3 mL of the culture was then inoculated into 300 mL of fresh LB medium for another 12 h cultivation at 37 °C and 220 rpm. Thereafter, the secondary seed was inoculated into 3 L fermentation medium in a 5 L reactor by 10% (*v*/*v*) inoculum size. The cultivation was kept at 37 °C and pH of 7.0, which was adjusted by adding 25% NH_4_OH or lactic acid. The original ventilation and impeller speeds were set as 1 *v*/*v*·min and 400 rpm, respectively. Upon observing an increase in dissolved oxygen (DO) levels, the feeding medium solution was added at a rate of 15 mL/h, maintaining the DO level at 30% by adjusting the cascading impeller speed to fluctuate between 400 and 850 rpm concurrently. Fermentation samples were taken at 12 h intervals for testing. The fermentation medium contained 10 g/L sucrose, 30 g/L yeast extract powder (FM902), 8 g/L NaCl, 3 g/L KH_2_PO_4_, 1 g/L MgCl_2_, 0.3 g/L CaCl_2_·2H_2_O, 0.2 g/L MnSO_4_·2H_2_O, 0.02 g/L FeSO_4_·7H_2_O, and 0.02 g/L ZnSO_4_·7H_2_O. The feeding solution of fed-batch contained 400 g/L sucrose and 100 g/L FM902.

### 2.5. Pululanase Assay

Pullulanase activity was measured as in our previous reports [28]. The total reaction volume was 500 μL, which contained 50 μL 5% (*w*/*v*) pullulan substrates, 50 μL pullulanase solution, and 400 μL phosphate buffer (pH 6.0, 50 mM). Following incubation in a water bath at 60 °C for 30 min, the reaction was stopped by the addition of 500 μL DNS solution and subsequent boiling for 10 min. The resulting reducing sugar was measured at 540 nm absorbance value. Glucose served as the standard for quantification of reducing sugar. A single unit (U) of pullulanase activity was defined as enzyme amount required to generate 1 μmol of reducing sugar per minute.

### 2.6. Determination of Pullulanase Expression Using SDS-PAGE

The pullulanase expression levels, both extracellular and intracellular, were determined by SDS-PAGE method according to our previous work [28]. Total volume of 10 μL sample solution containing 4 × loading buffer was loaded onto a 5% stacking gel and then running in a 12% separating gel. After electrophoresis, the separated proteins were stained with Coomassie Brilliant Blue G250.

### 2.7. Trehalose Production Process from Maltodextrin in Multi-Enzyme Catalyzed System

The neutral enzymes maltooligosyltrehalose synthase (MTSase) and maltooligosyltrehalose trehalohydrolase (MTHase) were produced according to our previous work [8]. The neutral pullulanase PulA3E produced in this work was used to enhance trehalose yield by co-catalyzing with more than two enzymes. The dosages of MTSase and MTHase were set as 100 U/g maltodextrin and 40 U/g maltodextrin based on our previous work [8]. The dosages of pullulanses were set as 10, 20, and 40 U/g maltodextrin, respectively. The concentration of substrate maltodextrin was 200 g/L. The reaction was performed at pH 6.0 and 60 °C for 48 h. Boil the reaction solution for 20 min to stop the reaction. A commercial pullulanase (Promozyme^®^ D2 purchased from Sigma-Aldrich, St. Louis, MO, USA) was set as control with the same gradient dosage as PulA3E. The trehalose product’s quantity was assessed using the HPLC method as outlined in our prior research [8].

## 3. Results and Discussion

### 3.1. Multiple Copies Genome Integration of PulA3E in B. subtilis 0127

Here, we used a single gene silencing strain *B. subtilis* 0127, which enhances the atypical secretion by silencing the encoding gene (*ytxE*) of a macromolecular protein transporter [21] as the expression host for the neutral pullulanase PulA3E. PulA3E was a mutant involving triple-sites R503E/I506E/H507E substitution of the neutral pullulanase PulA with enhanced extracellular expression in *B. subtilis* through an atypical secretion pathway [14]. Firstly, the encoding gene of PulA3E was successfully integrated into the *amyE* site of the *B. subtilis* 0127 genome using the CRISPR/cas9 method (Figure 1). The transcript of gene *pulA3E* was guided by promoter Pcry3A, as shown in Figure 1. On this basis, another *pulA3E* gene and tandem *pulA3E* genes were integrated into the *ytxE* site of the genome under the promoter Pcry3A, respectively (Figure 2A,B). The extracellular activities of these recombinant strains showed that two-copy strain PB could produce remarkably higher extracellular activity than that of one-copy strain PA during the whole process of fermentation (Figure 2C). However, the three-copy strain PB-2 showed relatively lower extracellular activity than that of PB (Figure 2C). It properly indicated that the tandem expression of double genes at one integration site is not a good strategy for gene overexpression in *B. subtilis*. Additionally, as the SDS-PAGE results show (Appendix A), the strains PB-2 and PB made similar amounts of extracellular expression of PulA3E. That is probably because PB-2 made a relatively more misfolded protein of PulA3E than PB. And for the same reason, compared with 48 h fermentation, the PulA3E protein yields were increased, but the extracellular activities declined after 96 h fermentation (Appendix A).

Thus, on the basis of strain PB, the third copy of *pluA3E* was integrated into the *ytrF* site to construct recombinant strain PC (Figure 3A). Thereafter, the fourth copy of *pulA3E* was integrated into the *trpP* site and *nprB* site of the genome of PC to construct recombinant strains PD and PE, respectively (Figure 3B,C). The results showed that the extracellular activity increased with the increase in copy numbers of *pluA3E* except PE, which integrated the fourth copy of *pluA3E* in the *nprB* site (Figure 3D). Additionally, at this time, the extracellular protein expression amount of PulA3E was consistent with the production trends of extracellular enzyme activities (Figure 3D and Appendix A). It means that the relatively lower extracellular activity produced by PE was not caused by the misfolding of PulA3E but probably due to the inappropriate integration sites for pulA3E. In this study, except for the integration sites *amyE* and *nprB*, the integration sites *ytxE*, *ytrF*, and *trpP* were selected mainly based on the results of our previous work. In the previous study, we verified that silencing the three genes separately could remarkably enhance the extracellular expression of an amylase guided by the N-terminal domain CBM68, which was derived from PulA through the atypical protein secretion pathway in *B. subtilis* [21]. All of the three genes encode large molecular transporters of *B. subtilis* [21]. Therefore, we selected the three sites to integrate *pulA3E*, which not only increased the copy number but also enhanced the PulA-type atypical protein secretion pathway by knocking out all three genes at the same time. Additionally, as shown in the results, the integration of *pulA3E* in the *trpP* site made the most remarkable increase in its extracellular expression (Figure 3D and Appendix A). From the above results, we verified again that knocking out the key large molecular transporters *ytxE* (0127), *ytrF* (0059), and *trpP* (4127) [21], which had been found in our previous work was indeed useful to enhance the PulA-type atypical secretion expression of foreign proteins in *B. subtilis*.

### 3.2. Improving Extracellular Expression of PulA3E by Overexpressing Specific Transporters

In our previous work, some key large molecular transporters such as *ytnA* (0046), *comEA* (0572), *spoIIQ* (2143), and *yvbW* (3038) were found to be essential to the PulA-type atypical protein secretion pathway [21]. When the key encoding gene was knocked out, the foreign protein secretion guided by CBM68 was remarkably suppressed [21]. On this basis, we overexpressed *ytnA*, *comEA*, *spoIIQ*, and *yvbW,* respectively, by a high-copy plasmid to further enhance the extracellular expression of PulA3E in the recombinant strain PD in this work. The results showed that during 48 h fermentation before the activities reached the highest level, the overexpression of each transporter made higher extracellular activity and lower intracellular activity compared with those of the control strain PD (Figure 4). However, unlike other genes, the overexpression of *comEA* also showed relatively higher intracellular activity compared with that of PD at 48 h fermentation (Figure 4B). This is probably because the overexpression of *comEA* greatly improved the secretion efficiency of PulA3E (Figure 4A), which could enhance the overall expression rate of *pulA3E* in *B. subtilis*. The protein expression level of PulA3E both in and out of the cells corresponded well to enzyme activity before 48 h fermentation (Appendix A). However, after 48 h fermentation, the extracellular protein expression of PulA3E kept increasing (Appendix A), but the extracellular activity declined (Figure 4A), probably due to the misfolding of PulA3E or some metabolites inhibiting the enzyme activity. Additionally, the relatively highest increase in the extracellular secretion of PulA3E by the overexpression of *comEA* was consistent with the results of our previous work [21]. Thus, *comEA* could be identified as the most important regulatory gene in the PulA-type atypical secretion pathway in *B. subtilis* so far. Based on the above results, the two key genes (*comEA* and *yvbW*), which had relatively higher effects on the secretory expression of PulA3E, were selected for tandem expression in one plasmid to further enhance its extracellular expression. As shown in Figure 5, the recombinant strain PD-ARY, which overexpressed both *comEA* and *yvbW*, produced the highest extracellular activity of 345.1 U/mL with 96 h of fermentation. It was 4.5 times higher than that of the control strain. It showed a little higher than that of the strain which overexpressed *comEA* at 96 h fermentation; however, at 84 h of fermentation, both the overexpressed *comEA* and *yvbW* increased twofold in extracellular activity than the overexpressed *comEA* alone (Figure 5A). This indicated that the simultaneously overexpressed *comEA* and *yvbW* was more likely to have contributed to the enhancement of the extracellular secretion efficiency.

### 3.3. Fermentation of the Recombinant Strains in a 5 L Reactor

In order to detect the extracellular production level of PulA3E by the recombinant strains constructed in this work, the four-copy genome integration strain PD and the secretion-enhanced strain PD-ARY were selected for further scale-up fermentation in a 5 L reactor. As shown in Figure 6A, the extracellular activity produced by PD was up to 1767.1 U/mL at 54 h of fermentation, while the extracellular activity remained no significant change from 54 h to 72 h of fermentation. However, strain PD-ARY produced comparable extracellular enzyme activity (1751.9 U/mL) with that of strain PD at 54 h of fermentation, but the extracellular activity still maintained growth from 54 h to 78 h of fermentation (Figure 6B). It reached the highest extracellular activity of 2465.1 U/mL at 78 h of fermentation (Figure 6B). Thus, it is concluded that the enhanced PulA-type atypical secretion pathway, by overexpressing *comEA* and *yvbW,* could remarkably improve the extracellular expression level of PulA3E. This strategy could provide an effective method for other proteins’ extracellular expression through the PulA-type atypical secretion pathway in *B. subtilis*.

### 3.4. Enhanced Trehalose Production by Adding Neutral Pullulanase PulA3E

Based on the high extracellular yield of the neutral pullulanase PulA3E, it was used to improve the trehalose conversion rate by co-catalyzing with the neutral enzymes maltooligosyltrehalose synthase (MTSase) and maltooligosyltrehalose trehalohydrolase (MTHase) at pH 6.0 and 60 °C. The concentration of substrate maltodextrin was 200 g/L. The dosages of MTSase and MTHase were 100 U/g maltodextrin and 40 U/g maltodextrin, respectively. The dosages of PulA3E and the commercial pullulanase (D2) were set as 10, 20, and 40 U/g maltodextrin, respectively. After 48 h catalytic reaction by the mixed enzymes, the highest trehalose production was up to 166 g/L with a conversion rate of 83% at the PulA3E’s dosage of 40 U/g maltodextrin (Figure 7A). However, when the dosage of D2 was 40 U/g maltodextrin, the trehalose production was only 105 g/L (Figure 7B). Additionally, the highest trehalose production was only 122 g/L with the addition of D2 with the dosage of 20 U/g maltodextrin (Figure 7B). This is most likely because the commercial pullulanase D2 was a type II pullulanase also possessing hydrolytic activity against 1,4 glucoside bonds; the higher dosage of pullulanase would produce more glucose and maltose products, which could not be further catalyzed by MTSase led to a relatively lower trehalose conversion rate (Figure 7B). In conclusion, the neutral pullulanase PulA3E, which could increase the trehalose conversion rate by more than 80%, has good potential for industrial application in the current neutral double-enzyme catalyzed method of the trehalose preparation process [8].

## 4. Conclusions

In this work, three encoding genes of the key large molecular transporters *ytxE* (0127), *ytrF* (0059), and *trpP* (4127), which negatively regulated the PulA-type atypical secretion expression of foreign proteins in *B. subtilis*, were selected as the genome integration sites of *pulA3E*. The engineered strain PD containing four copies of *pulA3E* at the three sites and the *amyE* site in the genome of *B. subtilis* showed high extracellular pullulanase activity of 1767.1 U/mL after 54 h fermentation in a 5 L bioreactor. On this basis, *comEA* and *yvbW* were selected from the four positive regulation genes (*ytnA*, *comEA*, *spoIIQ*, and *yvbW*) to further enhance the PulA-type atypical secretion of PulA3E by overexpressing the two genes. The corresponding engineered strain PD-ARY, which simultaneously overexpressed *comEA* and *yvbW,* produced the highest extracellular pullulanase activity of 2465.1 U/mL at 78 h fermentation in a 5 L bioreactor. Additionally, the neutral pullulanase PulA3E with a high extracellular expression level in *B. subtilis* was used to increase the yield of trehalose by co-catalyzing with MTSase and MTHase at pH 6.0 and 60 °C. When the supplement dosage of PulA3E was 40 U/g maltodextrin, the trehalose yield was up to 166 g/L, which was 1.58 times higher than that supplemented for the commercial pullulanase with the same dosage. Therefore, both the neutral pullulanase PulA3E and its industrial production strain PD-ARY showed good potential for industrial application.

## Figures and Tables

**Figure 1 bioengineering-11-00661-f001:**
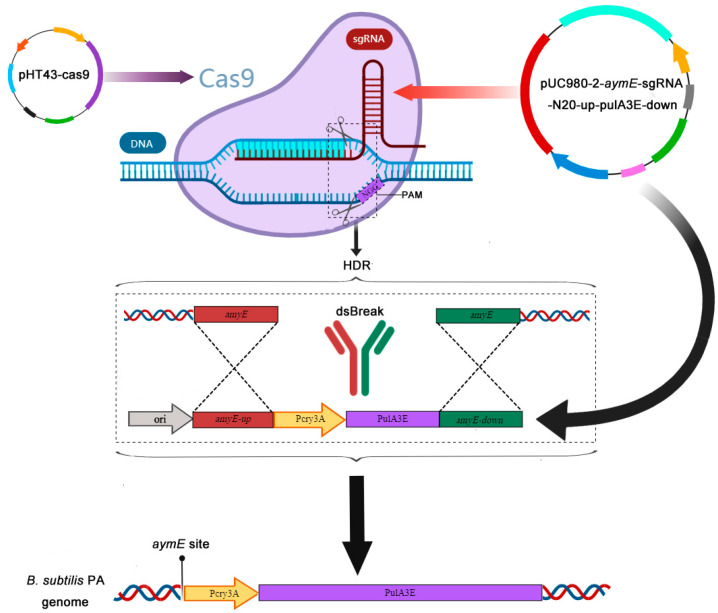
Schematic diagram of genome integration at *amyE* site by CRISPR/cas9 method.

**Figure 2 bioengineering-11-00661-f002:**
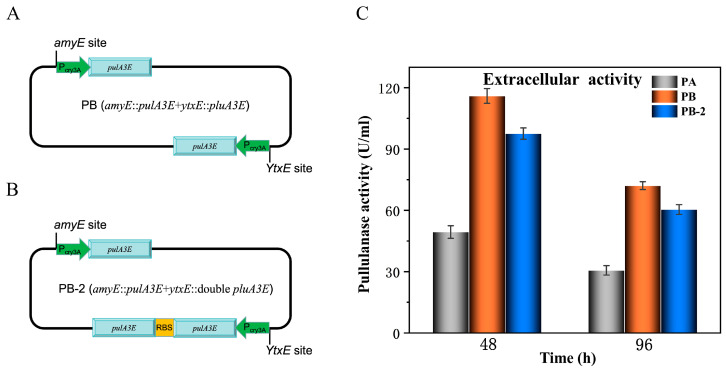
Schematic map of genomes of the double-site integration strains PB (**A**) and PB-2 (**B**) and the extracellular pullulanase activities of the strains (**C**). (**C**): The error bars depict the standard deviations derived from the mean values of three replicates.

**Figure 3 bioengineering-11-00661-f003:**
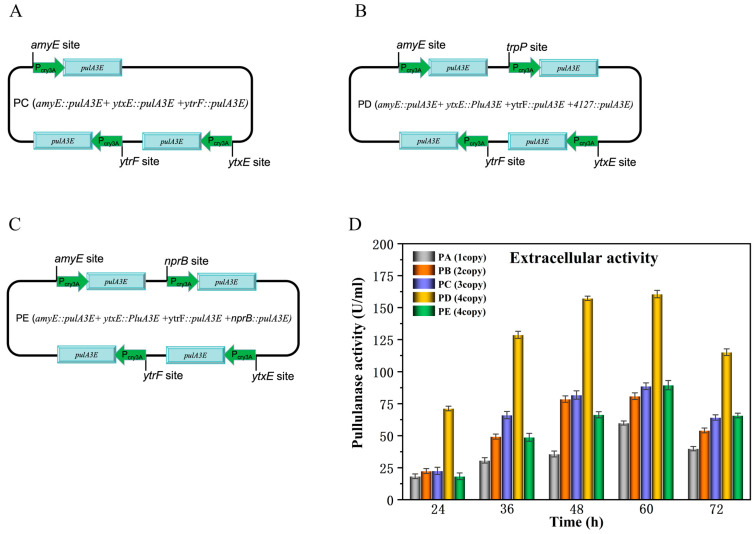
Schematic map of genomes of the multi-site integration strains PC, PD, and PE (**A**–**C**) and the extracellular pullulanase activities of the strains (**D**). (**D**): The error bars depict the standard deviations derived from the mean values of three replicates.

**Figure 4 bioengineering-11-00661-f004:**
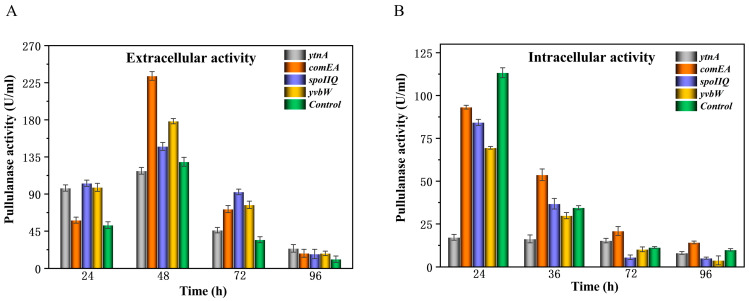
Effects of single gene overexpression of the key transporters for the PulA-type atypical secretion pathway on the extracellular pullulanase activities (**A**) and the intracellular pullulanase activities (**B**) of the engineered strains. The error bars depict the standard deviations derived from the mean values of three replicates.

**Figure 5 bioengineering-11-00661-f005:**
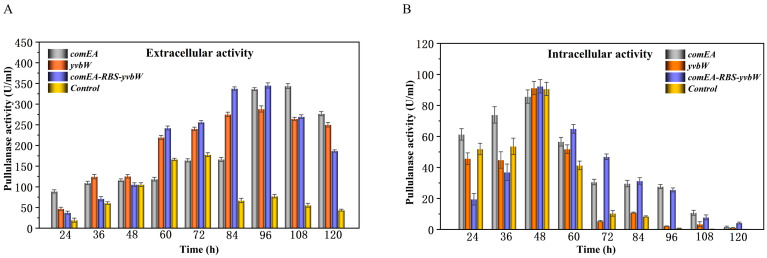
Effects of tandem expression of *comEA* and *yvbW* on the extracellular pullulanase activities (**A**) and the intracellular pullulanase activities (**B**) of the engineered strains. The error bars depict the standard deviations derived from the mean values of three replicates.

**Figure 6 bioengineering-11-00661-f006:**
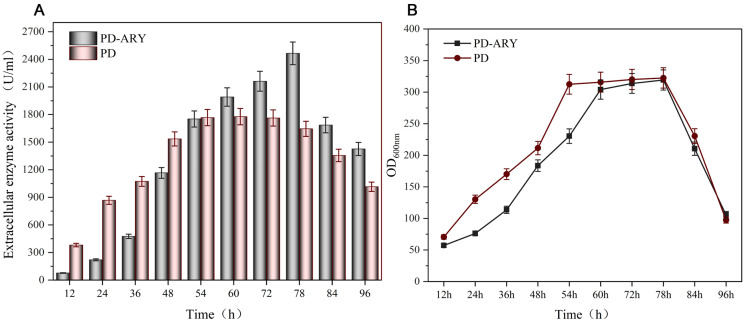
Scale-up fermentation in a 5 L reactor of the engineered strains PD and PD-ARY. (**A**) Extracellular enzyme activities of the recombinant strains; (**B**) growth profiles of the recombinant strains. The error bars depict the standard deviations derived from the mean values of three replicates.

**Figure 7 bioengineering-11-00661-f007:**
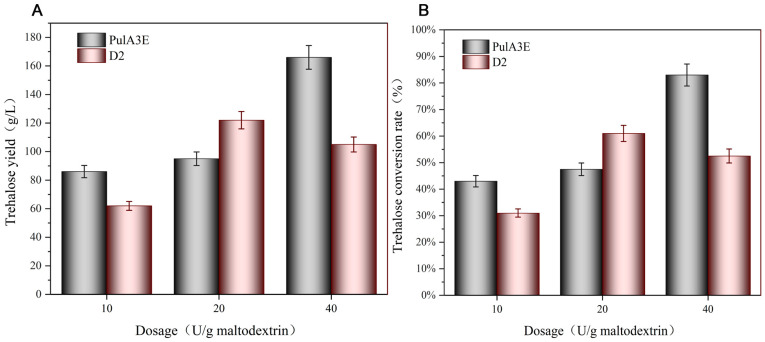
Effects of pullulanases PulA3E and D2 on trehalose production in the multi-enzymatic system. (**A**) Effects of pullulanase on the trehalose yield; (**B**) effects of pullulanase on the trehalose conversion rate. The error bars depict the standard deviations derived from the mean values of three replicates.

## Data Availability

All data generated or analyzed during this study are included in this published article and its Appendix A. All further data will be provided by the corresponding author at any time upon request.

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
