# Peer review of "Extracellular Overexpression of a Neutral Pullulanase in Bacillus subtilis through Multiple Copy Genome Integration and Atypical Secretion Pathway Enhancement"

_bioengineering, 2024, doi:10.3390/bioengineering11070661_

Round 1

Reviewer 1 Report

Comments and Suggestions for Authors

Extracellular overexpression of a neutral pullulanase in Bacillus subtilis through multiple copies genome integration and atypical secretion pathway enhancement

The m/s describes two different approaches, multiple copies genome integration to increase neutral pullulanase expression level and fermentation stability and second is to increase the secretory expression of PulA3E.The strain PD (four-copy genome integration) showed the highest extracellular pullulanase activity. The methodologies used to develop strains were proper.

The m/s needs some modifications as suggested.

Presently, industrial production of trehalose is by the two- enzyme method, using maltooligosyltrehalose synthase (EC 5.4. 99.15) and subsequently using maltooligosyltrehalose trehalohydrolase (EC 3.2. 1.141), with starch as the substrate. In the present m/s 3 enzyme approach is suggested.

The specific queries are:

1. What is the advantage of the 3 enzyme approach?

2. Instead of neutral pullulanase can one use isoamylase which has debranching activity too?

3. In the present case which is the limiting enzyme in the production of trehalose?

4. Pullulanase assay. The resulting reducing sugar was measured at 540 nm absorbance value. One pullulanase unit was defined as the amount of enzyme that produces 1 μmol reducing sugar per minute. What is the std. used? It should be per ml per min.

5. I could not find Fig.6A, B and 7A,B. (Line 258- 278)

6. Some typographical/grammer errors are:

Line 21. two strategies were employed to increase the production of a novel neutral pullulanase mutant, PulA3E, in B. subtilis.

Line 217. When the key encoding gene was knocking out, change to knocked out

Line 218. At the basis change to On the basis.

Comments on the Quality of English Language

--

Author Response

The m/s describes two different approaches, multiple copies genome integration to increase neutral pullulanase expression level and fermentation stability and second is to increase the secretory expression of PulA3E.The strain PD (four-copy genome integration) showed the highest extracellular pullulanase activity. The methodologies used to develop strains were proper.

The m/s needs some modifications as suggested.

Presently, industrial production of trehalose is by the two- enzyme method, using maltooligosyltrehalose synthase (EC 5.4. 99.15) and subsequently using maltooligosyltrehalose trehalohydrolase (EC 3.2. 1.141), with starch as the substrate. In the present m/s 3 enzyme approach is suggested.

The specific queries are:

  1. What is the advantage of the 3 enzyme approach?

Response: First of all, we are very grateful for the reviewer's positive comments. Actually, in the current industrial production of trehalose, the addition of pullulanase through the two-enzyme method is commonly practiced to enhance the conversion rate of trehalose from starch or maltodextrins. However, it is important to note that commercial pullulanases typically possess an acidic nature, rendering them unsuitable for the optimum reaction pH required by the neutral enzymes (MTSase and MTHase) that are widely employed in the current industrial production of trehalose. Therefore, the key novelty of this work lies not in using three enzymes to produce trehalose but in developing a new neutral pullulanase that is better suited for the current industrial production process compared to the commercially available acidic pullulanases. The relevant information has been added in the introduction section of the revised manuscript. (Line 44-51)

  1. Instead of neutral pullulanase can one use isoamylase which has debranching activity too?

Response: In theory, it is possible, but unlike isoamylase, pullulanase can break down the smallest unit of branched chains, maximizing the utilization of starch raw materials. Although isoamylase can hydrolyze the α-1,6 glycosidic bonds at branch points, it cannot hydrolyze side branches composed of 2 to 3 glucose residues.

  1. In the present case which is the limiting enzyme in the production of trehalose?

Response: Generally, for the production of trehalose, maltooligosyltrehalose synthase is the limiting enzyme. However, in the present case, we used the optimum dosages and ratio of the two enzymes according to the results of our previous report (Line 162-164). At the basis, we focus on revealing the effects of the pullulanse addition on the trehalose production in the multi-enzymatic system.

  1. Pullulanase assay. The resulting reducing sugar was measured at 540 nm absorbance value. One pullulanase unit was defined as the amount of enzyme that produces 1 μmol reducing sugar per minute. What is the std. used? It should be per ml per min.

Response: Glucose was used as the standard. The enzyme activity definition is consistent with other reports in this field. Because one pullululanase unit was defined as U not U/ml here. However, for the reviewers' comments, we have made appropriate additions and clarifications in section 2.5. (Line 149-151)

  1. I could not find Fig.6A, B and 7A,B. (Line 258- 278)

Response: We are so sorry for such omission. The Figure 6 and Figure 7 have been provided. (Line 281-285 and Line 306-310)

  1. Some typographical/grammer errors are:

Line 21. two strategies were employed to increase the production of a novel neutral pullulanase mutant, PulA3E, in B. subtilis.

Line 217. When the key encoding gene was knocking out, change to knocked out

Line 218. At the basis change to On the basis.

Response: The whole text has been carefully checked and modified for the typographical/grammer errors. (Line 21-22, Line 217, Line 218, Line 179, and Line 198, and so on)

Reviewer 2 Report

Comments and Suggestions for Authors

The authors have optimized expression of a neutral pullulanase from Anoxybacillus in the host Bacillus subtilis, modifying both copy number of the integrated pullulanase gene and the Bacillus secretion pathways to obtain the best outcome. The utility of the neutral pullulanase in a trehalose production reaction was also demonstrated and compared to that of a commercial standard enzyme.

The work is presented clearly, thoroughly and in a logical manner, with sufficient experimental detail. I have some minor comments that could be addressed by the authors.

1) In comparing the extracellular enzyme activity of strains PD and PD-ARY (Figure 6) the authors comment that at 54 hours the strains are effectively equal, but that the PD-ARY strain continues to produce more activity up to 78 hours while the PD strain has plateaued. The PD-ARY data presented (Figure 6B) extends out to 96 hours while the PD data (Figure 6A) stops at 72  hours. If possible it would be my preference to see the data for the PD strain out to 96 hours as well so that the visual comparison of the two datasets is more direct. Ideally both would be plotted on the same y-axis scaling as well for the same reason.

2) As with Figure 6, the data presented in Figure 7 would be more easily interpreted if the scaling of both graphs in panels A and B were identical. The different scaling of both yield and conversion rate between the panels makes direct comparison difficult.

Comments on the Quality of English Language

Firstly I commend the authors for their efforts in preparing the manuscript; the English language usage in the manuscript is fully understandable and interpretable which is an improvement on many papers I've seen from native English speakers! There are, however, several grammatical and word-use issues that should be revised. I would recommend that a native English speaking editor be employed to correct the language.

Author Response

The authors have optimized expression of a neutral pullulanase from Anoxybacillus in the host Bacillus subtilis, modifying both copy number of the integrated pullulanase gene and the Bacillus secretion pathways to obtain the best outcome. The utility of the neutral pullulanase in a trehalose production reaction was also demonstrated and compared to that of a commercial standard enzyme.

The work is presented clearly, thoroughly and in a logical manner, with sufficient experimental detail. I have some minor comments that could be addressed by the authors.

1) In comparing the extracellular enzyme activity of strains PD and PD-ARY (Figure 6) the authors comment that at 54 hours the strains are effectively equal, but that the PD-ARY strain continues to produce more activity up to 78 hours while the PD strain has plateaued. The PD-ARY data presented (Figure 6B) extends out to 96 hours while the PD data (Figure 6A) stops at 72 hours. If possible it would be my preference to see the data for the PD strain out to 96 hours as well so that the visual comparison of the two datasets is more direct. Ideally both would be plotted on the same y-axis scaling as well for the same reason.

Response: Many thanks for the reviewer’s kindly strong affirmation of our work. The data for the PD strain out to 96 hours have been added in Figure 6. Moreover, in order to make the visual comparison of the two datasets more directly, the Figure 6 A and B have been rearranged according to the reviewer’s good suggestion. (Line 281-285)

2) As with Figure 6, the data presented in Figure 7 would be more easily interpreted if the scaling of both graphs in panels A and B were identical. The different scaling of both yield and conversion rate between the panels makes direct comparison difficult.

Response: As with Figure 6, the Figure 7 A and B have also been rearranged to make direct comparison easy. (Line 306-310)